# N-Doped Modified Graphene/Fe_2_O_3_ Nanocomposites as High-Performance Anode Material for Sodium Ion Storage

**DOI:** 10.3390/nano9121770

**Published:** 2019-12-12

**Authors:** Yaowu Chen, Zhu Guo, Bangquan Jian, Cheng Zheng, Haiyan Zhang

**Affiliations:** Material and Energy School, Guangdong University of Technology, Guangzhou 510006, China; 1515707972627@163.com (Y.C.); ZhuFu19990@163.com (Z.G.); jianbangquan@163.com (B.J.); hyzhang@gdut.edu.cn (H.Z.)

**Keywords:** sodium-ion storage, graphene/Fe_2_O_3_ composite, nitrogen-doping modified graphene, high-performance anode material

## Abstract

Sodium-ion storage devices have received widespread attention because of their abundant sodium resources, low cost and high energy density, which verges on lithium-ion storage devices. Electrochemical redox reactions of metal oxides offer a new approach to construct high-capacity anodes for sodium-ion storage devices. However, the poor rate performance, low Coulombic efficiency, and undesirable cycle stability of the redox conversion anodes remain a huge challenge for the practical application of sodium ion energy storage devices due to sluggish kinetics and irreversible structural change of most conversion anodes during cycling. Herein, a nitrogen-doping graphene/Fe_2_O_3_ (N-GF-300) composite material was successfully prepared as a sodium-ion storage anode for sodium ion batteries and sodium ion supercapacitors through a water bath and an annealing process, where Fe_2_O_3_ nanoparticles with a homogenous size of about 30 nm were uniformly anchored on the graphene nanosheets. The nitrogen-doping graphene structure enhanced the connection between Fe_2_O_3_ nanoparticles with graphene nanosheets to improve electrical conductivity and buffer the volume change of the material for high capacity and stable cycle performance. The N-GF-300 anode material delivered a high reversible discharge capacity of 638 mAh g^−1^ at a current density of 0.1 A g^−1^ and retained 428.3 mAh g^−1^ at 0.5 A g^−1^ after 100 cycles, indicating a strong cyclability of the SIBs. The asymmetrical N-GF-300//graphene SIC exhibited a high energy density and power density with 58 Wh kg^−1^ at 1365 W kg^−1^ in organic solution. The experimental results from this work clearly illustrate that the nitrogen-doping graphene/Fe_2_O_3_ composite material N-GF-300 is a potential feasibility for sodium-ion storage devices, which further reveals that the nitrogen doping approach is an effective technique for modifying carbon matrix composites for high reaction kinetics during cycles in sodium-ion storage devices and even other electrochemical storage devices.

## 1. Introduction

With the rapid development of portable mobile electronic devices and electric vehicles, the importance of high-performance electrical energy storage (EES) devices is becoming increasingly prominent. The EES devices based on sodium ion such as sodium ion batteries (SIBs) and sodium ion supercapacitors (SICs) have attracted great attention due to their low cost, the rich abundance of sodium resource, and their high energy density, which approaches that of lithium-ion energy storage devices [1,2,3]. However, there are still some obstacles hindering the practical application of sodium ion energy storage devices, such as poor rate performance, low coulombic efficiency, and undesirable cycle stability. The foremost reason is the bigger radius of Na^+^ (0.102 nm) compared to Li^+^ (0.076 nm), which leads to low reaction kinetics for Na^+^ insertion/extraction from the anode materials [4,5,6]. Meanwhile, excellent anode materials should have suitable microscopic internal structures that can accommodate the lager Na^+^ and enable reversible insertion/extraction electrochemical behavior, and should also be compatible with electrolytes, cost-efficient, and easy to prepare. Therefore, the exploitation of prominent anode materials, synchronously coupling with optimum electrolytes, is the main challenge for sodium ion energy storage devices.

For Na^+^-storage negative electrode, alloy-based materials such as transition metal oxides (TMOs) and transition metal sulfides (TMSs) have been attracting increased interest because of their tailored morphology and high capacity. Among them, Fe_2_O_3_ is considered to be one of the most promising candidates due to their high theoretical specific capacity (1007 mAh g^−1^), abundance in earth, easy availability, low cost, and environmental friendliness [7,8]. Compared with bulk electrode materials, nanoscaled materials provide larger active surface area and shorter ion transport distances, which can significantly improve ion diffusion efficiency and enhance rate capability [9]. Valvo et al. investigated nanostructured Fe_2_O_3_ in composite-coated electrodes for the first time as a potentially viable negative electrode for SIBs with a capacity of 250 mAh g^−1^ after 60 cycles at 130 mA g^−1^ [10]. Nevertheless, iron oxides in nano size suffer from low electron conductivity and the defect of volume expansion and contraction during the charge/discharge processes, finally leading to low capacity and poor cycling performance [11,12,13]. To resolve this issue of volume change, a good strategy is to coat conductive carbon materials with Fe_2_O_3_ to form multicomponent composite materials. Then, graphene is considered to be the preferred carbon material because of its special two-dimensional structure, the excellent electrical conductivity and large theoretical specific surface area (2630 m^2^ g^−1^) [1,14]. The graphene nanosheets can provide enough active surface to anchor nano-Fe_2_O_3_ particles and offer an extremely conductive framework to promote electron transport [15]. Thus, a series of nanostructured Fe_2_O_3_/graphene composites, for example Fe_2_O_3_@GNS (graphene nanosheet), Fe_2_O_3_/r-GONRAs (reduced graphene oxide nanorod arrays), Fe_2_O_3_/N-GNS (nitrogen-doped graphene nanosheet), a-Fe_2_O_3_/pBC-N (nitrogen-doped pyrolyzed bacterial cellulose), γ-Fe_2_O_3_/PCF (porous carbon fibers), Fe_2_O_3_@C@MoS_2_@C and so on, were synthesized by various methods as anode electrode materials for SIBs with diverse electrochemical properties listing in Table 1. Among these, Li et al. synthesized Fe_2_O_3_/rGO nanocomposites, delivered 500 mAh g^−1^ after 100 cycles at a current density of 50 mA g^−1^, which was attributed to the single crystal structure of the Fe_2_O_3_ particles, and could maintain the structural integrity of the electrode and the electrochemical reversibility of the conversion process [15]. Chen’s group prepared 3D (three-dimensional) porous γ-Fe_2_O_3_@Carbon nanocomposites with ultrasmall Fe_2_O_3_ nanoparticles, stable C matrix, and 3D porous structure, providing a capacity of 317 mAh g^−1^ even at a current density of 8 A g^−1^ [16]. 

In this paper, nitrogen atoms were introduced to fabricate N-doping graphene/Fe_2_O_3_ (GF) nanocomposites as Na^+^ storage anode materials for SIBs and SISs. Fe_2_O_3_ nanocrystalline particles with a size of about 50 nm are uniformly anchored onto graphene nanosheets to obtain the GF sample through a one-step water bath synthesis method. Then, in order to further reduce graphene oxide for high conductivity, GF-300 was prepared. Finally, nitrogen atoms were doped into the graphene nanosheets by a further heat treatment in ammonia gas, and the obtained sample was noted as N-GF-300. The nitrogen-doped graphene-based framework shown in Figure 1 resulted in some advantages in this work, as follows: 1) enhancing the Fe_2_O_3_-graphene binding through the large electron affinity of nitrogen; 2) improving the nanocomposites of electrical conductivity to accelerate charge transport; 3) increasing reactive sites to improve energy conversion efficiency; and 4) producing the defects into the graphene nanosheets for high capacity [27].

Benefitting from the superior structural advantages, the final sample of N-GF-300 was proved to have excellent Na^+^ storage properties as the anode material for SIBs and SICs. The relative SIBs delivered a reversible discharge capacity of 638 mAh g^−1^ at 0.1 A g^−1^ and retained 300 mAh g^−1^ at 0.5 A g^−1^ after 100 cycles. Meanwhile, the asymmetrical N-GF-300//graphene SIC (the N-GF-300 sample was as the anode and graphene as the cathode material) exhibited a high energy density and power density with 58 Wh kg^−1^ at 1365 W kg^−1^ in 1 M NaClO_4_ organic solution. Furthermore, the electrochemical impedance spectroscopy (EIS) was utilized to seriously analyze the reason nitrogen doping into N-GF-300 could enhance the electrochemical performance, and it was finally revealed that nitrogen doping into the carbon-based electrode material is an effective method to improve the electrochemical properties for electrode materials.

## 2. Experimental Section

### 2.1. Synthesis of the Material

Firstly, the graphene/Fe_2_O_3_ (GF) nanocomposite was synthesized by a water bath process shown in Figure 1. In a typical synthesis procedure, 100 mL FeCl_3_·6H_2_O (0.901 g) solution was slowly added to 60 mL graphene oxide (2 mg mL^−1^) dispersion solution in a 250 mL round-bottom flask. After stirring for 2.5 h, 0.096 mL hydrazine hydrate (85 wt.%) was added as a reductant to the above solution; then, the mixed solution was subjected to a water bath at 80 °C for 24 h. The whole low-temperature hydrothermal process was kept under stirring. The product of the low-temperature hydrothermal reaction was treated by a series of procedures like vacuum filtering, washing and freeze-drying, and then the sample GF nanocomposite could be obtained. The GF-300 sample was prepared by annealing GF at 300 °C for 2 h under an argon atmosphere. The N-GF-300 was finally obtained via heating GF-300 to dope nitrogen atoms at 600 °C for 2 h in an ammonia atmosphere.

### 2.2. Structural Characterization of the Material

The morphology of the prepared material was observed by using a Hitachi SU8010 field emission scanning electron microscope (FESEM) (Hitachi, Japan). High-resolution transmission electron microscope (HRTEM) and mapping of the material were obtained by using a JEM-2100F field emission transmission electron microscope (JEOL, Japan). The Raman spectrum was collected using a Horiba LabRAM HR800 Raman spectrometer (HORIBA, France) with a 532 nm laser wavelength under a grating of 1800 gr/mm. X-ray diffraction (XRD) was measured by using a Bruker D8 Advance X-ray diffractometer (BRUKER, Germany) with a copper target. To test the content of Fe_2_O_3_ in the composite, thermogravimetric analysis (TGA) was recorded on a Thermogravimetric and synchronous thermal-infrared spectroscopy (Mettler-Toledo, USA) combined system in air from 10 to 800 °C at a heating rate of 10 °C min^−1^. X-ray photoelectron spectroscopy (XPS) was conducted on PHI 5000 VersaProbe II in situ X-ray photoelectron spectrometer (Ulvac-Phi, Japan). The specific surface area and pore size distribution curves were measured by the Brunauer Emmett Teller (BET) and density functional theory (DFT) method on a Micromeritics ASAP 2460 system (MICROMERITICS, USA), respectively.

### 2.3. Electrochemical Measurements

The electrochemical properties of the electrode material were measured by assembling them into a button cell. The electrode was produced by placing the mixture, which consisted of 60% material, 30% superconducting carbon black, and 10% PTFE (polytetrafluoroethylene) binder, onto a nickel foil and drying at 120 °C for 10 h under vacuum. The button cell (2032 type) was assembled in an argon glove box with water and oxygen content below 0.1 using the previously prepared electrode material as the working electrode, sodium as the auxiliary electrode and the counter electrode. The electrolyte used was 1 M sodium trifluomethanesulfonate (NaCF_3_SO_3_) in diglyme = 100 vol%, 1M NaClO_4_ in EC (ethylene carbonate): DMC (dimethyl carbonate): EMC (ethyl methyl carbonate) = 1:1:1 vol% with 5% FEC and 1 M NaPF_6_ in EC:DMC:EMC = 1:1:1 vol% with 5% FEC. The galvanostatic discharge/charge test was performed on a Land cycler (Wuhan Kingnuo Electronic Co., Wuhan, China). Cyclic voltammetry (CV) was tested on an CHI660E (CH Instruments, Shanghai, China) electrochemical workstation at a certain scan rate between 0.005 V and 3 V. The electrochemical impedance spectroscopy (EIS) was measured on the CHI660E electrochemical workstation with an alternating current (AC) voltage of 5 mV amplitude from 100 kHz to 100 mHz. For sodium ion capacitor devices, the cathode was made with a mixture of 60% graphene, 30% superconducting carbon black, and 10% PTFE and then placed on foamed nickel. For the anode, it was the same as the anode material used in the above assembled sodium ion battery, followed by assembly into a device in a glove box. The electrolyte used was 1 M sodium perchlorate (NaClO_4_) in EC:DMC:EMC = 1:1:1 Vol% with 5% FEC. The assembled devices were tested for EIS, CV, galvanostatic charge and discharge (GCD), etc., and the test parameters are the same as before. The specific energy density and power density of the sodium ion capacitor were calculated by the following formula [28]:(1)$$E\;({\mathrm{Wh}\;\mathrm{kg}}^{-1})=\frac{0.5C\times \Delta V^{2}}{3.6}$$
(2)$$P\;({W\;\mathrm{kg}}^{-1})=\frac{E\times 3600}{\Delta t}$$
where *C* is the specific capacitances of the supercapacitor cell, △*V* (V) is the voltage window during discharge process after the IR drop, △*t* (s) is the discharge time, *E* (Wh kg^−1^) is the specific energy density and *P* (W kg^−1^) is the power density.

## 3. Results and Discussion

Figure 1 shows a sketch diagram of the synthesis process of the N-GF-300 nanocomposite through a water bath and an annealing process in NH_3_ atmosphere. The water bath method with low temperature is a feasible approach for synthesizing the gram level above for nanomaterials with high production and low energy consumption. 

The morphology of GF, GF-300, N-GF-300 was measured by FESEM, as shown in Figure 2. In the SEM image of N-GF-300 (Figure 2c), Fe_2_O_3_ nanoparticles with a homogenous size of about 30 nm were uniformly anchored on the graphene nanosheets, revealing a better connection between Fe_2_O_3_ nanoparticles with graphene nanosheets compared to GF and GF-300. The nanostructured Fe_2_O_3_ particles in the N-GF-300 composite are tightly coated by the N-doped graphene sheet matrix, which facilitates the sodium ions rapidly inserted and extracted to improve the electronic conductivity. Most importantly, in terms of size, Fe_2_O_3_ particles in N-GF-300 are smaller than those in GF and GF-300 because of the greater number of active sites and the heterogeneous nucleation caused by N-doped graphene sheets [29]. The isotherm diagrams in Figure 2d clearly show the type IV adsorption isotherm with a distinct hysteresis loop indicating the mesoporous structure of these materials. The annealing at 300 °C for GF-300 with lower specific surface area (SSA) (97.9 m^2^ g^−1^) revealed the removal of unstable microporous structures from GF (102.1 m^2^ g^−1^), and the N-doping process further improved the SSA to 102.1 m^2^ g^−1^ with richer mesopores. As a whole, the low SSA and abundant mesoporous structure of N-GF-300 can effectively reduce the side reactions of electrolyte and increase ion transport channels. The XRD patterns were measured to investigate the crystal structure of the as-prepared GF, GF-300 and N-GF-300 materials, and all characteristic peaks were in good agreement with the standard card of Fe_2_O_3_ (JCPDS card No. 33-0664) (Figure 2e). The distinct peak of graphene is inconspicuous from the XRD pattern, which is attributed to the low graphene sheet content and the high removal of oxygen inserted into the graphite layer by reduction [30,31]. Raman spectroscopy was conducted to characterize the structural features of the graphene in GF, GF-300 and N-GF-300 composites. As shown in the Raman spectra (Figure 2f), the two main peaks located at 1394 and 1592 cm^−1^ were attributed to be D and G band of graphene, respectively [32]. The front band (D band) was attributed to the graphite lattice vibration of the Raman-active E^2^_g2_ mode, while the back band (G band) was attributed to the A1g mode, which comes from certain types of disorders and defects in the graphite crystal structure. The I_D_/I_G_ (the intensity ratio of the D and G bands) ratios are presented in increasing order of GF, GF-300 and N-GF-300, and are 1.28, 1.37 and 1.38, respectively. The larger number of the I_D_/I_G_ demonstrates the more amorphous structure of the carbon-based materials [33]. Therefore, the N-GF-300 composite has a more disordered microstructure than GF-300 and GF, and this result indicates that the nitrogen atoms had been successfully doped and the defects had been produced in the structure of the graphene sheets in N-GF-300, changing the perfect six-carbon ring structure [34]. 

HRTEM and mapping images of N-GF-300 were measured in order to further study the microstructure and the distribution of chemical elements as shown in Figure 3a–d, respectively. In Figure 3a,b, the Fe_2_O_3_ and graphene phases can be clearly distinguished. In addition, in Figure 3c,d, the mapping of N-GF-300 confirmed that the elements of, C, O, Fe and N were uniformly distributed in the composite. The HRTEM image (Figure 3e) shows that lattice fringes of the composite can be observed with a pitch of (012). XPS measurements were performed to further understand the surface chemistry of N-GF-300 and reveal that the weight content of N is up to 3.66%, which is a moderate doping level. As shown in Figure 3g, the two major peaks of Fe 2p located at 711.2 and 724.9 eV correspond to Fe 2p_2/3_ and Fe 2p_1/2_, respectively, which is consistent with Fe_2_O_3_ previously reported [18,35]. More importantly, the XPS spectra manifest the strong interface interaction that exists between Fe_2_O_3_ and N-doping graphene nanosheets in the composite of N-GF-300. As seen from Figure 3f, the C1s spectrum of N-GF-300 is mainly composed of six kinds of bonds: C–C/C=C (284.6 eV), C–N (285.6 eV), C–O (286.5 eV), C=O (287.8 eV) and O=C–O (289.4 eV) [7,18,21,36]. The N1s spectrum is made up of three peaks in Figure 3h, located at 398.6, 399.8 and 401 eV, which correspond to pyridinic-N, pyrrolic-N and graphitic-N, respectively [37,38].

The contents of three N-doping types as pyridinic-N, pyrrolic-N and graphitic-N (Figure 4b) in N-GF-300, GF-300, GF samples were compared in Figure 4a. The peak strength and peak area of pyridinic-N in N-GF-300 is much bigger than that of GF-300 and GF, and the result indicates that the nitrogen doping carried out by ammonia gas was beneficial in increasing the content of pyridinic-N. Notably, pyridinic-N was proved to enhance the conduction of electrons and facilitate diffusion of sodium ions [39]; therefore, N-GF-300 is more beneficial to improving the electrochemical performance for SIBs and SISs.

The electrochemical performance of N-GF-300 as the anode material of SIBs and SISs was expressed by CV, GCD and EIS. In the SIBs, three different electrolytes, including 1.0 M NaClO_4_, 1.0 M NaPF_6_ in EC:DMC:EMC = 1:1:1 vol% with 5.0% FEC, respectively, and 1.0 M NaCF_3_SO_3_ in diglyme were investigated by GCD, as shown in Figure 5a. Compared to 1.0 M NaClO_4_ and 1.0 M NaPF_6_ in EC:DMC:EMC = 1:1:1 vol% with 5.0% FEC, the SIBs with the ether-based electrolyte of 1.0 M NaCF_3_SO_3_ in diglyme delivered the highest discharge capacity of 378.4 mAh g^−1^ at a current of 0.5 A g^−1^. The reason for this is attributed a special solid electrolyte interphase (SEI) formed in the ether-based electrolyte [40], and the details of this will be discussed in our future work. Therefore, the studies on the electrochemical performance of SIBs carried out later were based on the ether electrolytes. Figure 5b shows the first three cycles of the CV curve for the N-GF-300 electrode measured at a scan rate of 0.05 mVs^−1^ at a range of 0.005–3 V. According to previous reports, the insertion and extraction of sodium ions in Fe_2_O_3_ is based on the following reaction [5,10,16,17]:(3)$$Fe_{2}O_{3}+6Na^{+}+6e^{-}↔2Fe^{0}+3Na_{2}O$$

In the first CV curve from Figure 5b, there are two cathode peaks appearing at 0.5 and 0.89 V, respectively, due to the formation of the SEI film and the produce of Na_x_Fe_2_O_3_ from the insertion of sodium ions into the Fe_2_O_3_. In the second CV curve, the cathode peaks at 0.63 and 0.94 V are accompanied by anode peaks at 0.6 and 1.6 V, which correspond to the reversible redox reaction of Fe^3+^ ↔Fe^2+^ and Fe^2+^ ↔Fe^0^ [5,16]. Figure 5c provides a comparison of the CV curves of the three electrode materials of GF, GF-300 and N-GF-300 at a scan rate of 1 mV s^−1^. Obviously, the CV curve area of the N-GF-300 material is larger than the other two, corresponding to the highest capacitance of 703 F g^−1^ compared to 532.5 F g^−1^ of GF-300 and 300.5 F g^−1^ of GF. Figure 5d is the rate performance of the three samples at different current densities. The discharge specific capacities of N-GF-300 at 0.1, 0.2, 0.5, 1, and 2 A g^−1^ are 638.4, 565.6, 464.9, 380.7 and 297.4 mAh g^−1^, respectively. When the current density returned to 0.1 and 0.2 A g^−1^, the discharge specific capacities of N-GF-300 could also return to the start level with 632.2 and 520.2 mAh g^−1^, respectively. In general, compared to GF and GF-300, the rate performance of N-GF-300 is superior to GF and GF-300, and more obvious at the higher the current density, which is attributed to the increasing charge transport of N-GF-300 by doping nitrogen atoms into the graphene sheets. The N-GF-300 electrode exhibits stable cycling performance and high Coulombic efficiency, as shown in Figure 5e. After 100 cycles at a current density of 0.5 A g^−1^, the discharge capacity of N-GF-300 could be maintained at up to 88.8%, while only 53% and 50.5% of the discharge capacity could be retained for GF-300 and GF respectively.

Electrochemical impedance spectroscopy (EIS) was utilized to further discover the internal energy storage mechanism of N-GF-300 as the anodes in SIBs. Figure 5f provides the EIS measurements of three composite electrode materials. The Nyquist diagram consists of a semicircle that intersects the graph and the accompanying line portion. The semicircular portion (in the high frequency region) corresponds to the electron transfer limiting process, and the straight portion (low frequency region) represents the diffusion limiting process. In the faster electron transfer process, the AC impedance spectrum contains only the straight portion, while the slower electron transfer process is characterized by a large semicircular region. The diameter of the semicircle is equal to the electron transfer resistance (R_ct_). The intercept of the semicircle on the real axis (Z_re_) to the electrolyte resistance R_e_ [41,42]. According to the circuit diagram [43], the fitting results are shown in Table 2. The solution resistance of N-GF-300 and GF-300 is similar to that of GF for using the same electrolyte, but the electron transfer resistance of N-GF-300 (R_ct_ = 18.24 Ω) is obviously smaller than that of GF (R_ct_ = 52.31 Ω) and GF-300 (R_ct_ = 40.17 Ω), indicating that the doping of N atoms is advantageous to enhance the electrical conductivity and electrochemical activity of N-GF-300 during the electrochemical process for the anode of SIBs [34].

Except for the anode material of the SIBs, N-GF-300 is also a highly appropriate anode material for SICs. In this work, an asymmetric SIC (Graphene//N-GF-300) comprised of graphene as the cathode material and N-GF-300 as the anode material was investigated by a series of electrochemical measurements, as shown in Figure 6, and a symmetric SIC N-GF-300//N-GF-300 was assembled for comparison.

The CV curves of the SICs Graphene//N-GF-300 and N-GF-300//N-GF-300 in Figure 6a maintained a typical quasi-rectangular shape. At a scan rate of 20 mV s^−1^, the specific capacitance of the symmetric SIC N-GF-300//N-GF-300 was 130 F g^−1^, while the Graphene//N-GF-300 supercapacitor had a much higher specific capacitance of 228 F g^−1^ due to the asymmetric configuration. Figure 6b shows the rate performance of the Graphene//N-GF-300 and N-GF-300//N-GF-300 capacitors at current densities of 1, 2, 5, 8, 10, 1 and 2 A g^−1^. The results show that the electrochemical performance of Graphene//N-GF-300 is significantly better than that of N-GF-300//N-GF-300. The energy density and power density of the SICs were calculated according to Equations (1) and (2), as shown in the Ragone plot (Figure 6c). The Graphene//N-GF-300 asymmetric capacitors exhibit higher energy density and power density compared to the N-GF-300//N-GF-300 symmetric capacitors. The energy density of the asymmetric capacitor reached 58 Wh kg^−1^ (1365 W kg^−1^) at a current density of 1 A g^−1^. Even with the current density reaching up to 10 A g^−1^, the energy density of the asymmetric capacitor Graphene//N-GF-300 still retained 28.1 Wh kg^−1^ (10,116 W kg^−1^).

Herein, the EIS measurement was used to further verify the superiority of the Graphene//N-GF-300 capacitor by analyzing internal resistance value of SICs. Figure 6d shows the Nyquist plots of the Graphene//N-GF-300 and N-GF-300//N-GF-300 devices, and the corresponding experimental impedance data and fitted data consisted of a loop in the high frequency region and a sloped line in the low frequency region. The equivalent circuit for fitting is R_e_(Q(R_ct_Z_w_))C_l_ and its schematic diagram is inset in Figure 6d. Here is a brief overview of the meaning of the five equivalent components: R_e_ is the equivalent series resistance, R_ct_ is the charge transfer resistance, Z_w_ is the Warburg impedance, Q is the constant phase element (CPE), and *C*_1_ is the finite capacitance. R_e_ includes the resistance of the electrolyte, the inherent resistance of the electrode material, and the resistance at the electrode material/collector interface. R_ct_ with respect to the diameter of the semicircle in the high frequency region indicates the dynamic resistance of charge transfer when the Faraday reaction occurs. Z_w_ is related to the slope of the 45° portion of the Nyquist plot in the intermediate frequency region, which is the impedance of the diffusion control process in the electrolyte. C_l_ is a finite capacitor. The electric double layer at the electrode/electrolyte boundary is usually equal to the electric double layer capacitor, but this deviates from the almost pure electrode layer due to the dispersion effect. The equivalent element denoted this part as CPE(Q). Its impedance is defined as:Z_Q_ = [Y_0_(jw)^n^]^−1^(4)
leading to
(5)$$Z\prime_{Q}=[Y_{0}(w{)}^{n}{]}^{-1}\cos\left(\frac{n\pi }{2}\right)$$
(6)$$Z{\prime \prime }_{Q}=[Y_{0}(w{)}^{n}{]}^{-1}\sin\left(\frac{n\pi }{2}\right)$$
where the dimension unit of Y_0_ is 1 S cm^−2^ s^−n^, which is a constant independent of frequency, and n is an exponent with the value between −1 and 1. When n = −1, the CPE is a pure inducer; when n = 0, the CPE is a pure resistor; when n = 1, the CPE is a pure capacitor; when 0 < |n|< 1, the CPE is Q [33]. Table 3 lists the value of each element in the equivalent circuit calculated by the complex nonlinear least-squares (CNLS) method. As can be seen from Table 3, Graphene//N-GF-300 is obviously better than N-GF-300//N-GF-300. Figure 6e shows the cycle performance of Graphene//N-GF-300 capacitors at current densities of 5 A g^−1^. In addition, the two Graphene//N-GF-300 capacitor devices in series can illuminate a red LED lamp (minimum operating voltage is 3 V), as shown in Figure 6f.

## 4. Conclusions

In summary, a nitrogen-doping graphene/Fe_2_O_3_ composite material N-GF-300 was successfully prepared as Na^+^ storage anode for SIBs and SICs through a water bath and an annealing process. The nitrogen-doping graphene-based framework in N-GF-300 enhanced Fe_2_O_3_-graphene binding through the high electron affinity of nitrogen to improve electrical conductivity and accelerate charge transport. Meanwhile, the reactive sites and defects increased in the graphene nanosheets during the nitrogen atom doping process, which could improve energy conversion efficiency at high capacity. Finally, the excellent electrochemical performance of the relative SIBs was indicated, with high capacity, good rate capability and stable cycling performance. A reversible discharge capacity of 638 mAh g^−1^ was delivered at the current density of 0.1 A g^−1^, and 428.3 mAh g^−1^ was retained at 0.5 A g^−1^ after 100 cycles. The asymmetrical N-GF-300//graphene SIC exhibited a high energy density and power density with 58 Wh kg^−1^ at 1365 W kg^−1^ in organic solution. 

## Figures and Tables

**Figure 1 nanomaterials-09-01770-f001:**
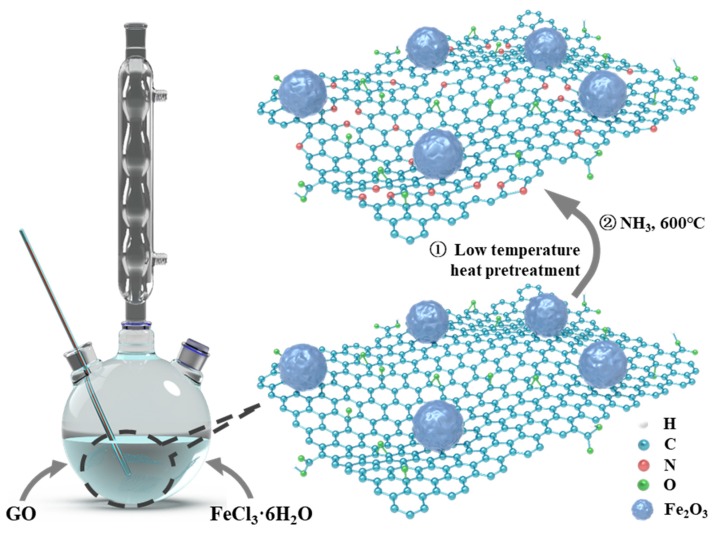
N-GF-300 synthesis diagram, wherein white: H atom; blue: C atom; red: N atom; green: O atom; blue purple: Fe_2_O_3_.

**Figure 2 nanomaterials-09-01770-f002:**
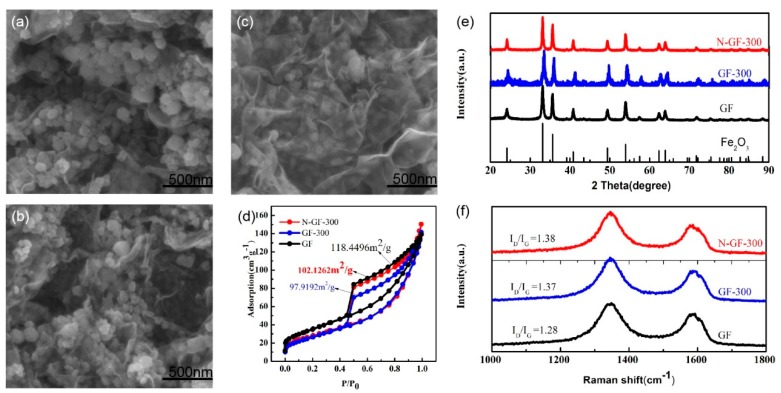
FESEM images of (**a**) GF; FESEM images of (**b**) GF-300; FESEM image of (**c**) N-GF-300; (**d**) Nitrogen adsorption/desorption isotherms; (**e**) XRD patterns of GF, GF-300 and N-GF-300; (**f**) Raman spectra of GF, GF-300 and N-GF-300.

**Figure 3 nanomaterials-09-01770-f003:**
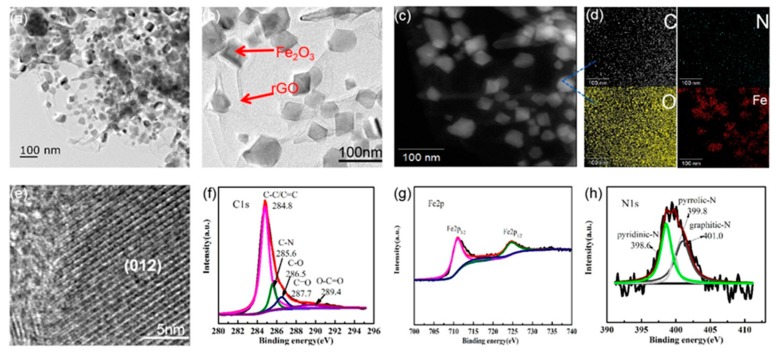
TEM image indicating a Fe_2_O_3_ sphere covered by graphene: (**a**) N-GF-300; TEM image indicating a Fe_2_O_3_ sphere covered by graphene: (**b**) N-GF-300; (**c**,**d**) Distribution of C, N, O and Fe in N-GF-300; (**e**) HRTEM image showing the lattice fringes; XPS spectra of N-GF-300: (**f**) C1s, (**g**) Fe2p, (**h**) N1s.

**Figure 4 nanomaterials-09-01770-f004:**
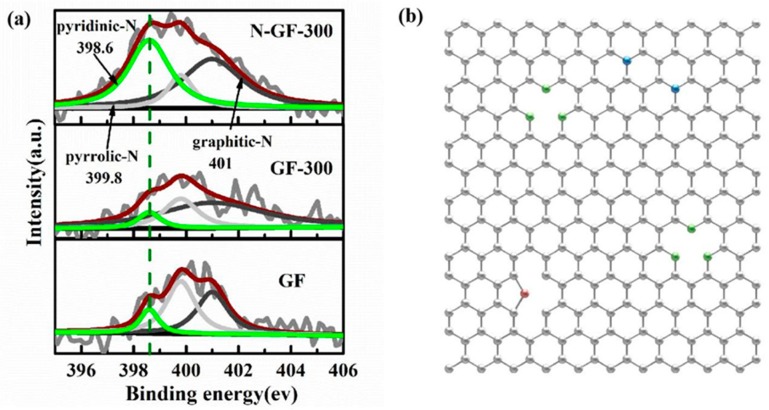
(**a**) N1s spectra of GF, GF-300 and N-GF-300, and (**b**) N-doped graphene framework, with corresponding configurations of N-containing functional groups, wherein green: pyridinic N, red: pyrrolic N, blue: graphitic N.

**Figure 5 nanomaterials-09-01770-f005:**
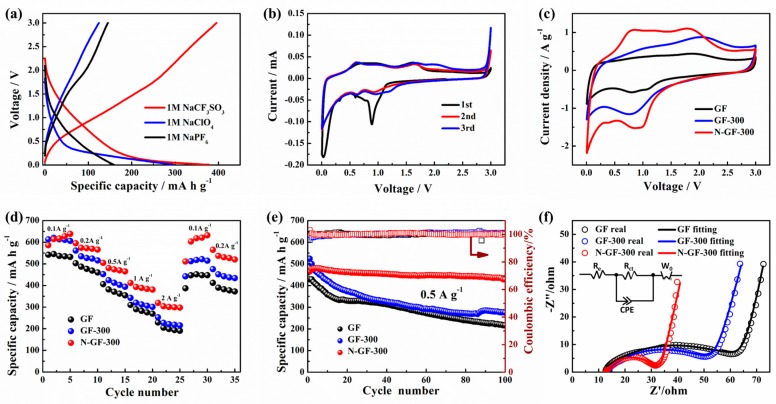
(**a**) Initial galvanostatic charge-discharge curves of different electrolytes for GF at 0.5A g^−1^ current density; (**b**) CV curves of the first three cycles for N-GF-300 between 0.005–3 V at scan rate of 0.05 mV s^−1^; (**c**) CV curves of GF, GF-300, N-GF-300 electrode at scan rate of 1 mV s^−1^; (**d**) Rate performance at different current densities; (**e**) cycle performance at 0.5 A g^−1^; (**f**) EIS of GF, GF-300 and N-GF-300 electrode.

**Figure 6 nanomaterials-09-01770-f006:**
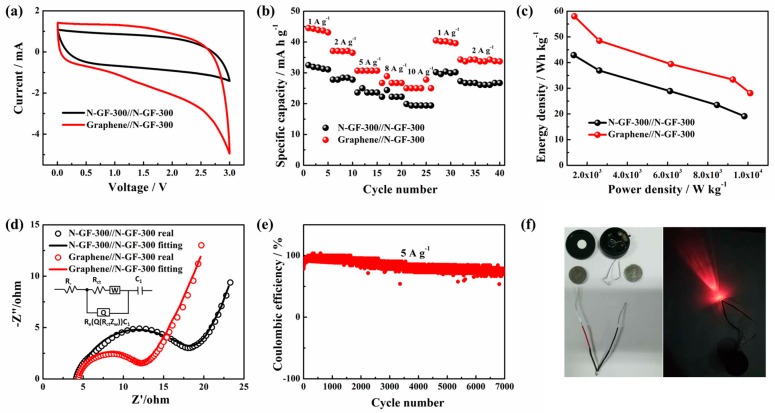
(**a**) CV curves of Graphene//N-GF-300, N-GF-300//N-GF-300 device at scan rate of 20 mV/s; (**b**) rate performance of Graphene//N-GF-300, N-GF-300//N-GF-300 device at different current densities; (**c**) Ragone plots of Graphene//N-GF-300, N-GF-300//N-GF-300 device at different current densities; (**d**) EIS of our prepared device-based Graphene//N-GF-300 and N-GF-300//N-GF-300; (**e**) cycling performance of Graphene//N-GF-300 device measured at a scan rate of 5 A g^−1^; (**f**) digital photograph of an electronic LED powered by two flexible devices in series.

**Table 1 nanomaterials-09-01770-t001:** Progress in the synthesis and electrochemical properties of Fe_2_O_3_ composites anode material for SIBs. (The definitions for the electrolyte abbreviations in Table 1 are listed here: EC (ethylene carbonate), DEC (diethyl carbonate), DMC (dimethyl carbonate), PC (propylene carbonate) and FEC (fluoroethylene carbonate))

Electrode Materials	Synthesis Method	Voltage Range (vs. N_a_/N_a_^+^)	Electrolyte	Cycling Data	Rate Capability	Ref.
Fe_2_O_3_@GNS	nanocasting technology	0.05–3	1M NaPF_6_ in EC:DEC=1:1	250/200th/0.5 A g^−1^	110/2 A g^−1^	[17]
Fe_2_O_3_/r-GONRAs	hydrothermal	0.01–3	1M NaPF_6_ in EC:DEC=1:1 with 5% FEC	~332/300th/0.2 A g^−1^	92/1.6 C	[5]
Fe_2_O_3_@GNS	ice bath	0.01–2.5	1M NaSO_3_CF_3_ in diglyme	110/500th/2 A g^−1^	194 /2 A g^−1^	[18]
Fe_2_O_3_/GO	freezing-dry	0.01–2.5	1M NaClO_4_ in EC:DMC=1:1	420/100th/0.1 C	90/10 C	[19]
Fe_2_O_3_-RGO	microwave assisted	0.005–3	1M NaClO_4_ in EC:PC=1:1	289/50th/0.05 A g^−1^	32.8/2 A g^−1^	[20]
Fe_2_O_3_/N-GNS	hydrothermal	0.01–3	1M NaClO_4_ in EC:DEC=1:1	306/50th/0.05 A g^−1^	132/1 A g^−1^	[7]
Fe_2_O_3_/rGO	solvothermal	0.01–3	1M LiPF_6_ in EC:DEC=1:1 with 5.0% FEC	~500/100th/0.05 A g^−1^	216/2 A g^−1^	[15]
3Dporousγ-Fe_2_O_3_@C	aerosol spray pyrolysis technique	0.04–3	1M NaClO_4_ in EC:DEC=1:1	358/1400th/2 A g^−1^	317/8 A g^−1^	[16]
a-Fe_2_O_3_/rGO	microwave hydrothermal	0.05–3	1M NaClO_4_ or LiPF_6_ in EC:DEC=1:1	310/150th/0.1 A g^−1^	77/2 A g^−1^	[21]
a-Fe_2_O_3_/pBC-N	in situ growth	0.01–3	1 M NaPF_6_ in EC:PC=1:1	408/350/0.1 A g^−1^	183/3 A g^−1^	[22]
γ-Fe_2_O_3_/PCF	-	0.01–3	1M NaSO_3_CF_3_ in tetraglyme	290/50th/0.1 C	230/2 C	[23]
Fe_2_O_3_@C@MoS_2_@C	hydrothermal	0.1–3	1M NaClO_4_ in EC:DEC=1:1	498/200/0.1 A g^−1^	150/2 A g^−1^	[24]
Fe_2_O_3_@NC	-	0.01–3	1mol/L NaClO_4_ in PC with 5% FEC	155.3/200th/4 A g^−1^	167.8/4 A g^−1^	[25]
α-Fe_2_O_3_@C/rGO	solvothermal	0.01–3	1mol/L NaClO_4_ in PC with 2% FEC	-	-	[26]
N-GF-300	water bath	0.005–3	1M NaSO_3_CF_3_ in diglyme	428.3/100th/0.5 A g^−1^	300/2 A g^−1^	this work

**Table 2 nanomaterials-09-01770-t002:** EIS Fitting results of N-GF-300, GF-300 and GF electrode before cycling.

	R_e_/Ω	R_ct_/Ω	CPE1-T/Ω	CPE1-P/Ω	W1-R/Ω	W1-T/Ω	W1-P/Ω
N-GF-300	12.71	18.24	0.0003998	0.6355	0.6355	0.3124	0.4290
GF-300	12.22	40.17	0.001024	0.4620	9.437	0.2949	0.4201
GF	10.25	52.31	0.0009205	0.4283	12.83	0.4646	0.4391

**Table 3 nanomaterials-09-01770-t003:** The value of *R_e_*, *C* (= *Q*, when *n* =1), *Q*(Y_0_, n), *R_ct_*, *Z_w_*(Y_0_) and *C*_1_ calculated by CNLS using the fitting EIS data based on the relative equivalent circuit.

Impedance Parameter	R_e_ (Ω)	R_ct_ (Ω)	Z_w_ (Y_0_)	C_1_ (F)	n	C (F, n = 1)
Graphene//N-GF-300	4.358	7.541	0.03987	0.04718	0.7537	0.000118
N-GF-300//NGF-300	4.142	14.09	0.027	0.0123	0.7348	0.0001265

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
