# Peer review of "N-Doped Modified Graphene/Fe2O3 Nanocomposites as High-Performance Anode Material for Sodium Ion Storage"

_nanomaterials, 2019, doi:10.3390/nano9121770_

Round 1

Reviewer 1 Report

This paper deals with the synthesis and characterization of novel nanocomposite materials for sodium ion batteries. In the introduction the authors justify their proposal and add 27 citations that provide further and relevant information concerning the field of study. Useful information is provided by a list on the synthesis and characterization of iron oxide nanocomposites as anodes for SIBs. With  a novel and simple innovative approach, as described in the experimental section, a novel material is described showing high performance behaviour as anode  electrode. The procedure adopted for the synthesis, structural and electrochemical characterization of the GF samples is described in this section 2., that would be more appelative if a few more details were included about the used techniques. In section 3., results and their discussion are reported. The structural and electrochemical performance of the prepared samples is well expressed in terms of clarity, presentation, deep scientific and technical approach. DC and AC measurements led to important data that confirm the superior capacity, rate capability, cycling stability, charge retention, and energy density / power density of the prepared N-GF-300 samples.In summary this is a good work, in  scientific and applied terms, deserving publication.

Author Response

Thank you very much for your good suggestion. We fully agree with your comment. In the revised manuscript, we have revised the corresponding descriptions.

Changes in the revised manuscript: In the section 2, more details about the GF sample were added in the revised manuscript.

The third paragraph in page 3 was changed from “Firstly, the GF nanocomposite was synthesized by a water bath process showed in Figure 1a. 100mL FeCl3·6H2O (0.901 g) solution was slowly added to 2 mg mL−1 GO dispersion solution. After stirring for 2.5 hours, the above solution was subjected to a water bath at 80℃ for 24 hours with continuous stirring. The solution after the reaction was subjected to suction filtration, and the sample obtained by suction filtration was freeze-dried for 24 hours to obtain a graphene/iron oxide (GF) nanocomposite. Then, the GF nanocomposite was annealed at 300 ℃ for 2 hours under an argon atmosphere to obtain GF-300. Finally, the N-GF-300 nanocomposite was got by heating GF-300 at 600 ℃ for 2 hours under ammonia atmosphere.” to “Firstly, the GF nanocomposite was synthesized by a water bath process showed in Figure 1a. In a typical synthesis procedure, 100mL FeCl3·6H2O (0.901 g) solution was slowly added to 60mL graphene oxide (2 mg mL−1) dispersion solution in a 250 mL round-bottom flask. After stirring for 2.5 hours, 0.096 mL hydrazine hydrate (85 wt.%) as a reductant was added into the above solution, then the mixed solution was subjected to a water bath at 80℃ for 24 hours. The whole low-temperature hydrothermal process was kept stirring. The product of the low-temperature hydrothermal reaction was treated by a series of procedures like vacuum filtering, washing and freeze-drying, then the sample graphene/Fe2O3 (GF) nanocomposite could be obtained. The GF-300 sample was prepared by annealing GF at 300 ℃ for 2 hours under an argon atmosphere. The N-GF-300 was finally got via heating GF-300 to dop nitrogen atoms at 600 ℃ for 2 hours in an ammonia atmosphere.”

Reviewer 2 Report

The manuscript entitled “N-doped Modified Graphene/Fe2O3 Nanocomposites 2 as High-Performance Anode Material for Sodium ion 3 Storage” describes the solution phase synthesis of Iron Oxide nanoparticles on nitrogen doped graphene for Sodium Ion electrochemical energy storage devices.

The manuscript provides a detailed discussion of the state of the literature and a thorough materials characterization and understanding of the synthesis process. The authors have demonstrated the applicability of their electrochemical devices to power practical systems (via an LED). Based on the thorough context, materials characterization, and electrochemical characterization, I believe the manuscript may be published with the following minor revisions.

Figure 1: Ball colours should be labelled on the figure as a legend (not just in the caption).

Figure 5: there is no (b) label

Figure 6: the resolution/size of the x-axis values should be increased, the superscript is nearly unreadable

Table 2: the significant figures used in the table seem arbitrary, there should be listed properly with an error value for context.

Methods: Raman should list the grating used for the measurement as this defines the spectral resolution
